# The Influence on Contaminant Bioavailability and Microbial Abundance of Lake Hongze by the South-to-North Water Diversion Project

**DOI:** 10.3390/ijerph16173068

**Published:** 2019-08-23

**Authors:** Yu Yao, Peifang Wang, Chao Wang

**Affiliations:** 1Key Laboratory of Integrated Regulation and Resource Development of Shallow Lakes, Ministry of Education, College of Environment, Hohai University, Xikang Road 1st, Nanjing 210098, China; 2School of Environment, Nanjing Normal University, Nanjing 210023, China

**Keywords:** South-to-North Water Diversion Project, DGT, next-generation high-throughput sequencing technique, sediments

## Abstract

The world famous South-to-North Water Transfer Project was built to alleviate serious water shortages in northern China. Considering that lake Hongze is an important freshwater lake in this region, analyzing the influence of water diversion on typical contaminant bioavailability and microbial abundance could aid in achieving a good overall understanding of hydrodynamic variation. Accordingly, in situ high-resolution measurements of diffusive gradients in thin films (DGT) and next-generation high-throughput sequencing were combined in order to survey Lake Hongze and determine the relationship between environmental factors and microbial communities. The DGT method effectively obtained more than the 85% of bioavailable concentrations of the corresponding contaminants; the results showed that labile P, S, Fe, As, and Hg concentrations were higher in areas influenced by water transfer. Moreover, the relative abundance and alpha diversity of the sampling sites distributed in the water transfer area differed significantly from other sites. The pH, conductivity, and labile Mn, As, and P were shown to be the primary environmental factors affecting the abundance and diversity of microbes. With the exception of bioturbation-affected sites controlled by labile Mn and pH, sites distributed in the water diversion area were most affected by As and conductivity, with little spatial discrepancy. Furthermore, site 2, with higher bioturbation abundance, and site 10, with stronger hydrodynamics, had low alpha diversity compared to the other sites. Consequently, the bioavailability of typical contaminants such as P, S, As, Hg, Fe, Mg, Cd, Pb, and Mn, as well as the diversity and abundance of microbial in the sites influenced by the water diversion, were significantly different to the other sites. Thus, the impacts of the South-to-North Water Transfer Project on participant lakes were non-negligible overall in the investigation.

## 1. Introduction

The freshwater ecosystem represents an important source of water for drinking and terrestrial plant irrigation, and water quality safety is a constant global environmental issue [1]. Due to the enormous difference between South and North China in terms of water quality and quantity, the South-to-North Water Diversion Project was implemented in 2013. This tremendous project could significantly affect the diverse and dynamic distribution of freshwater contamination in the immediate environment, while having nonnegligible impacts on biospheric biogeochemical cycling. The imported and exported water impel variations in the aquatic ecosystem and river hydrodynamics, creating the new environments with significantly altered physicochemical parameters. This phenomenon can, in turn, profoundly influence the aquatic microbial community distributions as well as their activities in the freshwater ecosystem [2]. Consequently, the contaminated fraction concentrations and microbial community distributions are fundamental to improving the overall understanding of the freshwater ecosystem variation influenced by the South-to-North Water Diversion Project. Moreover, mapping the spatial distribution of the contamination fractions and the microbials in different areas of freshwater is also vital for integrating studies of the influence of the South-to-North Water Diversion Project. 

Recent findings underscore the vital functions of sediments for allochthonous and autochthonous contaminant management [3]. Sediment acts as a regulator, trapping and accumulating exogenous contaminated fractions and subsequently releasing these fractions in different forms depending on the environmental physicochemical properties [4,5]. The sustained and substantial release of pollutants from sediment accounts for unrecoverable contamination in the overlying water over a relatively long period [6,7], as well as the delay in freshwater recovery despite exogenous pollutant import decline [8,9]. Furthermore, the surface sediment microbials are assumed to disperse with freshwater currents [10]. Considering the overlying water diverted with subsurface flow paths, microbial distribution together with the effect of the water diversion might be dependent upon surface sediment analysis accuracy. Related investigations have revealed that the gargantuan water diversion project could create a homogenization of the freshwater sediments as well as an increase or decline the contaminant concentrations over a long time scale [11]. However, investigations have revealed the hydrodynamics show profound differences in surface sediments in different areas [12]. Paradoxically, contaminants and microbial diversity might not concomitantly change. The water diversion induces undesirable ecological issues such as contaminants import and microbial diversity. Accordingly, the overall survey of sediment pollutant bioavailability and microbial diversity in water diversion-affected area is fundamental for the evaluation of the South-to-North Water Diversion Project.

Owing to complex environmental conditions and the enormous scale of the project, the development of robust and non-invasive methods for freshwater contamination evaluation is a key research priority. Compared to traditional automatic and grab sampling approaches, passive sampling techniques have many merits. Passive sampling can obtain in situ, continuous time-dependent contamination information, avoiding repeated spot sampling [12,13]. Among the extensive range of in situ passive sampling approaches, the diffusive gradients in thin films technique (DGT) has been successfully applied to predict various element concentrations in freshwater ecosystems [14,15]. Due to its ability to predict the contaminated pre-concentration, the DGT has the clear advantages in terms of a low detection limit even under extremely low concentrations or during relatively short deployment for target element accumulation [16,17,18]. The novel two-dimensional (2D) DGT technique can obtain contamination information at a submillimeter level for sediments, which can be combined with analyses such as laser ablation-inductively coupled plasma-mass spectrometry (LA-ICP-MS), routine 2D slicing, and the computer-imaging densitometry (CID) technique [19,20]. Combined DGT and CID could have extensive applications as a powerful in situ tool for collecting contamination data due to their quite rapid and inexpensive sample analysis characteristics [21,22]. This high-resolution 2D approach with DGT enables imaging of the heterogeneous distribution of target elements, especially their concentration gradients across the sediment–water interface (SWI), which is critical for target element diffusion/transmission between the sediment and the overlying water. Accordingly, the DGT technique, including 1D and 2D imaging, was used in this study to evaluate target element lability in sediments and their cycling in the freshwater ecosystem, which show large spatial variations in terms of geochemical characteristics [19]. 

There exists an intricate but inevitable relation between microbial and physicochemical properties. It is an truism that microbials can exist almost anywhere water. Microbial activities directly influence the physicochemical properties of water and have profound effects on biospheric biogeochemical cycling to some extent, given their highly dynamic and extremely diverse properties. Sediment microbial communities must undergo corresponding shifts in abundance at temporal and spatial scales due to environmental changes. The water diversion can clearly change the lake aquatic ecology and hydrology upstream, creating distinct environments with new physicochemical properties. In turn, the changes could also influence the sediment contaminants, especially environmentally sensitive contaminants such as Hg, As, P, and S. Thus, characterizing the biogeochemical processes involved in contaminant migration and transformation requires in situ measurements, independent of abiotic and biotic influences on the freshwater ecosystem [19]. Accordingly, the simultaneous application of dynamic and high-resolution technique, next-generation high-throughput sequencing is essential for the overall understanding of freshwater ecosystems. 

Consequently, in this study, the contaminated situation of Lake Hongze was comprehensively evaluated by in-situ DGT measurement and next-generation high-throughput sequencing in order to understand the migration and distribution of target elements and predict potential contamination trends. 

## 2. Materials and Methods

### 2.1. Site Description

Hongze lake is fourth largest freshwater lake as well as the largest plain type reservoir in China; it plays an important role in agricultural economy and water supply in Jiangsu province [5,23]. Fluvial landforms and tributary branching contribute to highly diverse water quality and microbial communities [19]. As shown in Figure 1, 18 sites, evenly distributed throughout Lake Hongze, were selected as the sampling sites. Sites 1, 2, 9, and 10 are located in the bay area of the lake, which is not obviously affected by hydrodynamic behavior (Figure 1). Sites 7, 8, 12, and 15–18 are located in clear water diversion-affected areas, in which contaminants might be driven by hydrodynamic fluctuations. Sites 3, 6, 11, 13, and 14 are all distributed in the inlet or outlet area of the lake. Moreover, the microbial community diversity between these sampling sites can reveal the influence of the South-to-North Water Transfer Project on microbial communities. The widespread distribution of aquaculture and exogenous pollution import resulted in the extremely serious eutrophication of Lake Hongze. In addition, Lake Hongze is a participant of the South-to-North Water Transfer Project, and the internal loading of its contaminants are also thought to be affected by water diversion. This hypothesis supported by previous ex situ investigations [24]. Owing to its significant economic and geographic position, as well as the complex contaminant migration, Lake Hongze might be a typical freshwater lake for this type of investigation.

### 2.2. Contaminant Analysis and Physicochemical Characteristics

To obtain the in situ contamination information, 1D and 2D DGT probes were used to obtain sediment contamination information for Hg, As, Fe, Mg, Cd, P, and S. Surface sediments (from the SWI to a 5-cm depth) were collected by a Peterson grab sampler for analysis of penetrating microelectrodes, high-throughput sequencing, and the determination of other physiochemical indices such as pH, dissolved oxygen concentration (DOC), moisture content, total N concentration, suspended solids, oxidation-reduction potential (ORP), and conductivity.

#### 2.2.1. In Situ Measurements

1D and 2D DGT probes were used to obtain sediment contamination information for Hg, As, Fe, Mg, Cd, P, and S. Surface sediments (from the SWI to a 5-cm depth) were collected by a Peterson grab sampler for analysis of penetrating microelectrodes, high-throughput sequencing, and other physiochemical indexes such as pH, dissolved oxygen concentration (DOC), moisture content, total N concentration, suspended solids, oxidation-reduction potential (ORP), and conductivity. Fe, Mg, Cd, Pb, and Mn concentrations, and AgI DGTs were used for S concentrations. The details of DGT preparation and assembly sequence are shown in the Appendix A. Briefly, to avoid accumulation on the surface of the resin gels in the sediment probes, the ZrO and AgI probes were deoxygenated by nitrogen for more than 24 h and then stored in oxygen-free 0.03 M NaNO_3_ solution before deployment in the sediments [9,22]. All DGT probes were deployed in the sediments of Lake Hongze using the new DGT release device shown in Appendix A. After retrieval, the resin gels of the 1D DGTs (3-mercaptopropyl functionalized silica gel-DGT probes, Chelex-100 DGT probes) were separated at 1-mm intervals, and two strips in combination were sequentially added to the centrifuge tube with the corresponding extractants. The protocol for analysis of mercaptopropyl functionalized silica gel-DGT and Chelex-100 DGT probes have been shown elsewhere in detail [15]. 

To obtain the masses of Hg, As, Fe, Mg, Cd, Pb, and Mn in the corresponding resin blanks, 15 standard DGT devices were immersed in the 4 L 0.01 M NaNO_3_ solutions for 24 h, after which the resin gels were eluted with the corresponding extractants; the resin gels could extract over than 85% corresponding fractions [9,18,20,21]. As Appendix A shows, the average masses of Hg, As, Fe, Mg, Cd, Pb, and Mn in the blank gels were determined as 0.37 ± 0.11 ng·cm^−2^, 4.2 ± 0.43 ng·cm^−2^, 20.58 ± 0.77 ng·cm^−2^, 7.9 ± 0.16 ng·cm^−2^, 6.1 ± 0.51 ng·cm^−2^, 3.9 ± 0.43 ng·cm^−2^, and 31.2 ± 0.79 ng·cm^−2^, respectively. The measurement of detection limit (MDL) was calculated as three times the standard deviation of the blank gel accumulation mass, which converted to concentrations of 0.018 μg·L^−1^, 0.204 μg·L^−1^, 0.73 μg·L^−1^, 1.002 μg·L^−1^, 0.384 μg·L^−1^, 0.297 μg·L^−1^, and 0.846 μg·L^−1^, respectively. The ZrO DGT and the AgI DGT, employed as a typical 2D DGT, obtained the labile P and S concentrations combined with computer-imaging densitometry (CID). After retrieval from the sediment cores, the surfaces of ZrO DGT and AgI DGT were rinsed with deionized water. Then, the resin gel was immediately removed from the devices and adhered water was adsorbed by filter paper. The resin gels (with the ZrOCl_2_, AgI particles settled on the downward-facing surface) were then scanned by a flat-bed scanner (HP G4010) at a 600-dpi resolution (pixel size 42 × 42 μm). Lastly, Image J processing software (version 1.48, National Institutes of Health, Baltimore, MD, USA) was used to convert the scanned images of the ZrO and AgI resin gels to grayscale. Similar to the 1D measurement, the standard S solution was applied to ensure consistency of the CID analysis [15,21]. The detailed process of 2D DGT for labile S and P are shown in Appendix A. The ZrO DGT and AgI DGT performances indicated that the grayscale values obtained by the scanner were controlled at a relative standard deviation (RSD) of <5%. Detailed calculations for accumulated masses of 2D measurement (labile P and S) and 1D measurement (Hg, As, Fe, Mg, Cd, Pb, and Mn) have been reported elsewhere [21,22,25]. The RSD between sample duplicates was less than 5%. The certified values of target elements in standard solutions were all within the range of experimental error (<5%). 

For the penetrating microelectrode analysis, sediment cores (11 cm in diameter and 50 cm in length) were collected using a gravity corer (Rigo Co.,Tokyo, Japan) from the 18 sampling sites. The pH, ORP, H_2_S, and DOC values were measured in the sediments using the corresponding penetrating microelectrode (OX 100, Unisense, Aarhus, Denmark). The detailed procedures are in accordance with the published methods of Lewandowski and Hupfer [26] and Zhang et al. [27]. 

#### 2.2.2. ExSitu Measurements

Collected sediment samples were frozen at −80 °C in the ultra-low temperature freezer, then freeze-dried by a vacuum freeze dryer (LABCONCO, Kansas City, MO, USA, B16-333-8811) and sieved with a 2-mm stainless steel mesh. The details of the ex situ measurements for DOC, moisture content, dissolved total nitrogen, suspended solids (SS), and total K and Na concentrations are shown in the Appendix A. For the measurement and analysis of microbial community composition and diversity, sediment samples were carefully collected and stored at −80 °C until DNA extraction. Genomic DNA was extracted using the E.Z.N.A.^®^ Tissue DNA kit (Omega Bio-tek, Norcross, GA, USA) according to the manufacturer’s instructions. The V4–V5 hypervariable regions of 16S rRNA were PCR-amplified from microbial genomic DNA using the following universal primers: V515F, 5’-GTGCCAGCMGCCGCGG-3’; V907R, 5’-CCGTCAATTCMTTTRAGTTT-3’. Purified amplicons were pooled in equimolar and paired-end sequenced (2 × 250) on an Illumina MiSeq platform according to standard protocols by Majorbio Bio-Pharm Technology Co., Ltd (Shanghai, China) [27,28]. The resulting sequences were screened and filtered for quality and length. Sequences with a length shorter than 50 bp with more than two primer mismatches were removed, and paired end reads were merged accordingly [29]. 

The 16S rRNA MiSeq sequencing data were operated with the modified pipelines of mothur and UPARSE (http://www.drive5.com/uparse/) [29]. The values obtained by the number of sequences belonging to specific taxon divided by the number of total sequences were used to represent the relative abundance (%) of individual taxa within each community. The alpha diversity (using Shannon, Chao1, adaptive communication environment (ACE), Faith’s phylogenetic diversity (PD), Simpson, and Dominance values) and the beta-diversity (Bray–Curtis distance) were determined based on randomly selected sequences (the fewest sequences among the samples) from the sediment samples. To compare the species diversity between different samples, the rarefaction curve was constructed by sampling for all sequences. Then, the R program was applied to analyze the microbial community structure at different taxonomic levels [30]. 

#### 2.2.3. Statistical Analyses

All statistical analyses were performed using the SPSS statistical package (Version 18.0 for Windows, Stanford University, San Francisco, CA, USA) and some were implemented by R packages (The University of Auckland, Auckland, Auckland Region, New Zealand). Specifically, the SPSS was applied for normality testing of the physiochemical properties as well as the DGT-measured labile concentrations at different depths/widths of sampling sites; then, the *T* test was applied to verify the difference. The relative abundances of specific lineages in different sampling sites were clustered by the R package g plots; the *F* and *T* test were also applied to verify the relative abundances of dominant lineages between the different sampling sites. Furthermore, the community dissimilarities among the different sampling sites in Lake Hongze were determined based on Bray-Curtis distances. Abiotic factors influencing the community were evaluated using a BioENV procedure, and canonical correspondence analysis (CCA) were used to select the best environmental variables for the microbial communities of all sampling sites in Lake Hongze [31].

## 3. Results

### 3.1. Physicochemical Characteristics

The concentrations of pH, ORP, dissolved oxygen (DO), electrical conductivity (EC), total nitrogen (TN), and SS, as well as the temperature in the overlying water, were measured by the portable water analyzers (HQ 30d, HACH, Loveland, CO, USA) and standard methods. The moisture content of surface sediments in different sampling sites were obtained following the standard procedure [13].

The physicochemical characteristics of the overlying water and sediments are summarized in Table 1. There were significant differences between the temperatures of the 18 sampling sites because of the different sampling sequence. To ensure accurate evaluation of the DGT technique, the diffusion coefficients of labile fractions were selected for the corresponding temperature of the sampling site. As shown in Table 1, the total N, P, and suspended substance concentrations in the overlying water of site 2 differed significantly from other sampling sites. Site 2 was located in the north of the lake, an area developed by the aquaculture industry. The extremely high density of benthic organisms may contribute to the abnormal concentrations of TN, TP, and SS in the overlying water. Paradoxically, the physicochemical indices of sites 2, 3, 10, and 11 distributed in the water diversion-affected area did not concomitantly change with the hydrodynamic increase. The most perverse phenomenon is that the DO concentrations of sites 2 and 3 were much lower compared to the other sampling sites. We speculated the water diversion did not play a significantly role due to the widely distributed aquaculture and the characteristic of the perched stream.

### 3.2. Labile Fraction Analysis

Lake Hongze has been classed as having eutrophication freshwater, considering the large-scale blue algae bloom breakout that has occurred every year since 2014 [5]. Previous reports have demonstrated that the eutrophication degree is not only related to labile P concentrations but is also controlled by the heterogeneity index [9,13]. The high heterogeneity of labile P in the sediment raises the potential blue algae hazard as well as labile P fractions migration intensity. Thus, ZrO-DGT was applied in this study to obtain the labile P contamination information. As Appendix A shows, labile P concentrations varied from 5 to 300 μg·L^−1^ along the sediment profiles of sampling sites in Lake Hongze. Sites 2, 3, 10, and 11 indicated higher P bioavailability than the others; their sediment profiles indicated significant horizontal heterogeneity, and there were localized high concentration areas at these sites. The extremely low density of macrobenthos and deposition of the blue algal bloom might explain the existence of the localized high concentration areas. Macrobenthos, such as chironomid larvae and *Corbicula fluminea*, could make the contaminant distribution more uniform and transport dissolved oxygen along the sediment profile. Sites 10 and 11, located in the inlet of the lake, had a low density of macrobenthos, which might result in the localized high concentration areas along the sediment profiles. Although sites 2 and 3 were located at the outlet of the lake, the macrobenthos density may not be particularly low because of the large-scale cultivation influence. The northwest of the lake has exhibited the most serious area blue algal bloom outbreaks. The existence of the localized high concentration areas in sites 2 and 3 might be explained by blue bloom deposition. The 2D distribution of labile S concentrations also demonstrates the corresponding labile P variation at different sampling sites. The labile S concentrations ranged from 8 to 138 μg·L^−1^ along the sediment profiles, and sites 2, 3, 5, 9, 10, and 11 showed higher bioavailability of labile S than other sampling sites. Furthermore, the labile S variation could explain the localized high concentration areas in the sediment profiles of labile P. As Appendix A shows, the labile S concentrations in the overlying water at sites 2, 3, 10, and 11 were not proportional to their sediment labile S concentrations. The DGT-measured S levels of sites 2 and 3 indicated high lability at the bottom of the sediment profiles; however, labile S concentrations in the overlying water were extremely low. There were clear differences between labile S concentrations in the sediment and the overlying water in sites 2, 3, 9, 10, and 11. We speculated the wind fluctuations and bioturbation might contribute to the black bloom deposition. As the Table 1 showed, the extremely low values of DO and ORP in the overlying water could also reflect the phenomenon of bloom dissolved oxygen consumption. The FeS is the culprit for the black algal bloom. 

Anthropogenic activities, i.e., fossil fuel combustion, smelting, and electroplating, have led to long-term, irreversible heavy metal contamination of the freshwater ecosystem. Among the heavy metals, mercury (Hg), arsenic (As), cadmium (Cd), lead (Pb), and zinc (Zn) are notorious due to their toxicity (International Agency for Research on Cancer, Geneva, Switzerland). A statement by the World Health Organization indicated that at least 50 million people face the threat of chronic poisoning by heavy metals [32]. As well as fractions with high toxicity to biota, metal fractions such as manganese (Mn), iron (Fe), magnesium (Mg), calcium (Ca), and copper (Cu) were also important for contamination evaluation. The mobility and concentration of different labile fractions were limited by different physicochemical indexes, and the contaminants concentrations could conversely apply to the physicochemical variations [30]. Accordingly, Chelex–DGT was applied to obtain Cd, Pb, Zn, Fe, Mn, Ca, Mg, and Cu concentrations, and 3-mercaptopropyl functionalized silica gel–DGT was used for Hg concentrations in this study [5]. All measured labile fraction concentrations exhibited significant variations along the sediment profiles of sampling sites, with labile As, Hg, Cd, Pb, Zn, Fe, Mn, Ca, Mg, and Cu concentrations ranging from 0.17 to 4.21 μg·L^−1^, 0.05 to 1.29 μg·L^−1^, 0.23 to 8.53 μg·L^−1^, 0.04 to 1.33 μg·L^−1^, 3.54 to 150.3 μg·L^−1^, 5.0 to 212.5 μg·L^−1^, 0.18 to 6.88 μg·L^−1^, 0.20 to 7.61 μg·L^−1^, 0.22 to 8.35 μg·L^−1^, and 0.04 to 1.52 μg·L^−1^, respectively (Appendix A and Figure 2). Except for labile Hg, all labile fractions showed a clear variation trend with increasing sediment depth. Gaseous Hg could enter the freshwater by atmospheric deposition; atmospheric transport of Hg accounts for at least 70% of total labile Hg in the freshwater ecosystem [33,34]. Moreover, wind fluctuations and benthic disturbance could also lead to labile Hg returning to the atmosphere [13]. This dynamic exchange process between the sediment-overlying water and the water-atmosphere resulted in the irregular variation of labile Hg between the sediments [5,19]. As shown in Appendix A, there was a clear valley or peak in the vicinity of the sediment–water interface for all measured fractions in the sediment profiles. Furthermore, a simultaneous change occurred in labile Fe-As and Fe-P concentrations between the sediments of different sampling sites. Meanwhile, an opposing effect between labile Cd and Pb concentrations was found in the sampling sediment of Lake Hongze. Similar to the 2D measurement of labile P and S (Appendix A), higher bioavailable concentrations of As and Hg were observed in sites 2, 3, 10, and 11 than in other sampling sites. Related investigations showed that higher valence of the Fe, Mn, Al, and Ca fractions are strongly adsorptive to anion fractions such as the AsO_4_^3−^, AsO_3_^3−^, and PO_4_^3−^ [11]. Meanwhile, the HgS as a percentage of Hg^−^ in the sediments of eutrophication lake was consistent and very high by other reports [15]. This phenomenon could also demonstrate the microbial functions of the labile Hg concentrations [13]. 

### 3.3. Microbial Community Composition and Diversity

To ensure that the full extent of microbial diversity was sufficient for investigation of the 18 sampling sites, rarefaction analysis was applied. 16S rRNA Miseq sequencing obtained a total of 63,006 operational taxonomic units (OTUs) from Lake Hongze. The proportion of total obtained OTUs differed significantly for different sampling sites, from 577 to 4281 OTUs. The ranges of Shannon, Chao1, ACE, Faith’s PD, Simpson, and Dominance values were 5.51–10.14, 586–6882, 593–7165, 90–384, 0.932–0.997, and 0.003–0.068, respectively. Among all the sampling sites, sites 2, 3, 10, and 11 showed significant advantages compared to other sites, indicated by Dominance values of 0.068, 0.017, 0.025, and 0.021, respectively (Table 2). Furthermore, to evaluate the difference between all sampling sites, the Bray–Curtis similarity metric was applied for the principal components analyses (PCA) (Appendix A). The results of the PCA revealed that sampling sites were highly concentrated, except for sites 2 and 10. The 18 sampling sites showed a significant difference in the Archaea and Bacteria composition of the sediment community by taxonomic classification (*T* test, all *p* < 0.05). As shown in Appendix A, Bacteria was dominant, accounting for 91.3–96.8% of the total classifiable 16s rRNA sequence, while a very small proportion of the classifiable sequence belonged to Archaea. Moreover, the dominance of Archaea varied significantly between sampling sites. The proportion of Aenigmarchaeota in the total identified sequence varied from 0.012% to 0.34% in the 18 sampling sites. Euryarchaeota showed an obvious advantage in sites 12 and 14, accounting for 3.8% and 1.4% of the 16s rRNA sequence classification, respectively. Similarly, Crenarchaeota accounted for 0.3% in site 6, 1.1% in site 10, and the majority of the 16s rRNA sequence in the corresponding sampling sites. In total, 47 bacterial phyla were detected in the sampling sites, displaying very different proportions of total phyla. Among them, Actinobacteria was the dominant bacteria, accounting for 36.2–78.6% across the 18 sampling sites, and Firmicutes, Chloroflexi, and Planctomycetes made up 3.8–18.3%, 0.6–14.3%, and 1.1–17.1% of total phyla, respectively. The results of the MiSeq sequencing revealed a large variation in the Bacteria/Archaea community in the 18 sampling sites. At the class level, there were 14 Archaea and 104 Bacteria classes (Appendix A). At the genus level (Appendix A), there were 33 Archaea and 715 Bacteria genera. *Pseudoxanthomonas*, *Desulfovibrio*, *Mahella*, and *Hydrogenoanaerobacterium* were detected as the primary members in 18 sampling sites, but their proportion of the total abundance was highly variable. As shown in Appendix A, the composition of the microbial community in site 2 was very different from other sites. *Desulfarculus* and *Desulfovibrio* were clearly the dominant genera, accounting for almost 10% and 31.5% of total genera in site 2, respectively, and indicating a much high proportion of the total bacterial abundance than other sites. They are both typical sulfate-reducing bacteria, and might contribute to the extremely high labile S concentrations in site 2. However, the extremely low proportions of *Caldithrix* and *Methylomicrobium* resulted in higher labile Fe and MeHg concentrations than at other sites. In as in site 2, site 10 also showed unique results, and the extremely high relative abundance of *Methylobacillus* in site 10 resulted in its high Hg bioavailability. 

## 4. Discussion

Contaminant distribution is assumed to reflect the combined effects of microbial and physicochemical properties. Moreover, variations in physicochemical properties could cause substantial variation in microbial activities and community composition, which might lead to selection of tolerant species and, in turn, alter physicochemical properties. The relationship between the microbial community, physicochemical properties, and contaminants was analyzed to provide an overall understanding of the Lake Hongze aquatic ecosystem. As Figure 3 shows, DOC, ORP, pH, Fe, Mn, P, Mg, and DO were the main factors influencing the microbial community structure of different sampling sites in Lake Hongze. These factors all play different roles: DOC, ORP, and pH had the greatest impact on microbial community structure variability. Moreover, all factors are closely correlated. Related studies have indicated that labile Fe and Mn concentrations might exert a dominant control on ORP variations, and that labile Mg concentrations are related to pH variations. Furthermore, environmental factors indicate a similar correlation with the microbial community structure at most sampling sites. However, the microbial community structure of sampling sites located at the inlet and outlet of the river differed significantly from other sampling sites. This phenomenon might be the result of the South-to-North Water Transfer Project, as sampling sites in the northern and southern areas of the lake were clearly affected by water transfer. Moreover, the microbial community structure of sampling sites affected by water transfer also indicated distinct characteristics. Sites 3 and 4 were clearly controlled by pH and Mg, sites 2, 10, and 11 showed a significant correlation with ORP and Fe variations, and sites 1 and 6 were affected by DOC variations. The characteristics of sampling sites affected by water transfer are noteworthy for further investigation. To determine the key environmental factors controlling microbial community structure, a simple linear correlation between DOC, ORP, pH, Simpson, and Faith’s PD is shown in Figure 4. pH was a primary factor controlling alpha diversity and was negatively correlated with Faith’s PD (R^2^ = 0.632, *p* <0.05) and Simpson parameters (R^2^ = 0.578, *p* <0.05). The DOC also indicated a negative correlation with Faith’s PD (R^2^ = 0.759, *p* <0.05) and Simpson (R^2^ = 0.804, *p* <0.05) parameters. The ORP showed a positive correlation with Faith’s PD and Simpson parameters; the coefficients were 0.7425 (*p* <0.05) and 0.6201 (*p* <0.05), respectively. These three environmental factors indicated a significant correlation with the alpha diversity of the microbial community in Lake Hongze. Due to rapid progression of the South-to-North Water Transfer Project, the environmental factors controlling microbial community diversity may undergo substantial changes, and these changes between environmental and microbial factors will significantly affect anthropic activities and health. 

## 5. Conclusions

To thoroughly understand the contamination of labile fractions and the microbial community structure in Lake Hongze, the in situ DGT and next-generation high-throughput DNA sequencing techniques were combined for an environmental evaluation of the lake. The contaminants distributed at the bioturbation influence area were found at clearly higher levels. However, the microbial abundance and diversity were conversely lower. The labile P, S, As, and Hg concentrations in areas that might be affected by water transfer were clearly higher or lower than other sites. Moreover, the microbial community diversity of sampling sites located in bioturbations and water diversion affected areas were distinctly lower compared with the other sites, as indicated by the relative abundance and alpha diversity of the microbes in different sites. Furthermore, the results of the CCA and simple linear analysis showed that environmental factors such as pH, DOC, and ORP could significantly influence the microbial community structure in the entire lake. However, sampling sites distributed in areas influenced by water transfer also showed unique characteristics. Some sites were controlled by changes in labile Mg concentrations and pH, and some sites were restrained by labile Fe/Mn concentrations and ORP variability. 

## Figures and Tables

**Figure 1 ijerph-16-03068-f001:**
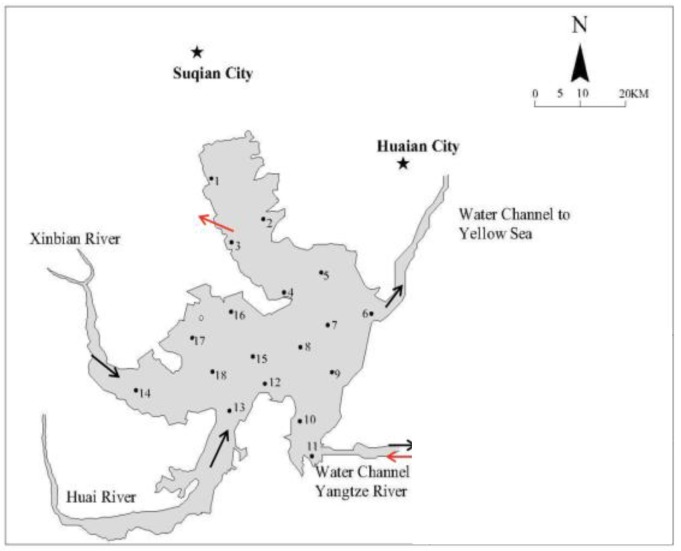
The distribution of the sampling sites in Lake Hongze. The black arrow represents the flow direction before the water transfer; the red arrow represents the flow direction after the water transfer.

**Figure 2 ijerph-16-03068-f002:**
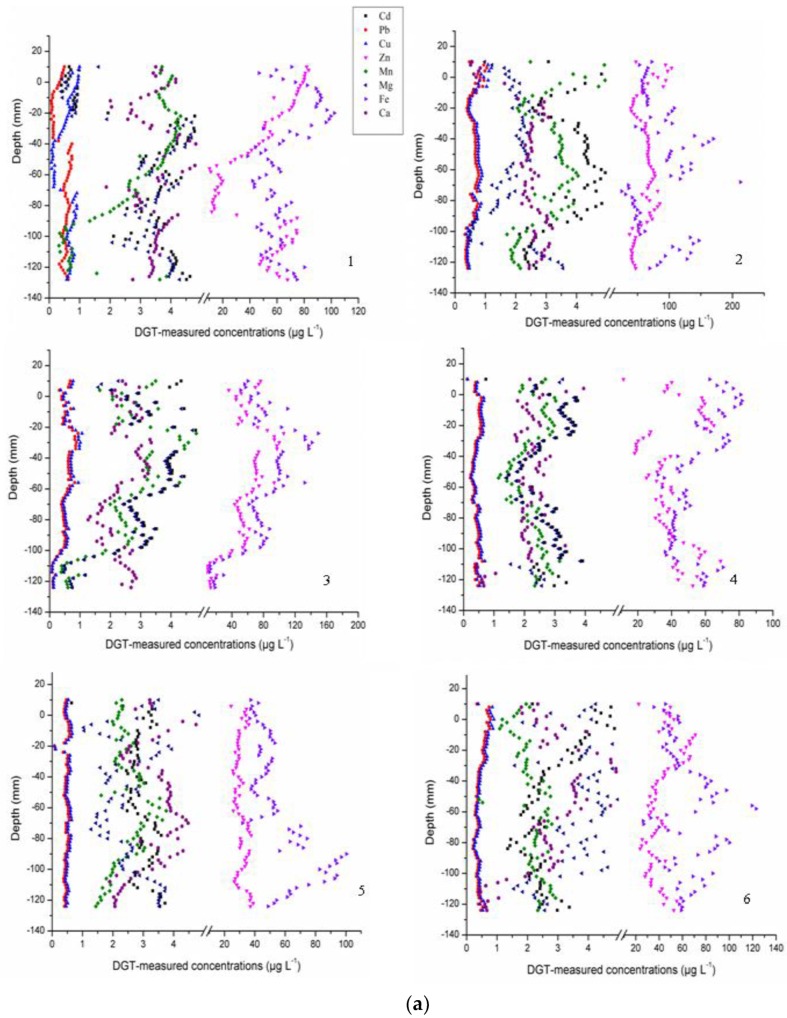
(**a**) The diffusive gradients in the thin films technique (DGT)-measured Cd, Pb, Zn, Fe, Mn, Ca, Mg, and Cu concentrations of sampling sites 1–6 in Lake Hongze. (**b**) The DGT-measured Cd, Pb, Zn, Fe, Mn, Ca, Mg, and Cu concentrations of sampling sites 7–12 in Lake Hongze. (**c**) The DGT-measured Cd, Pb, Zn, Fe, Mn, Ca, Mg, and Cu concentrations of sampling sites 13–18 in Lake Hongze.

**Figure 3 ijerph-16-03068-f003:**
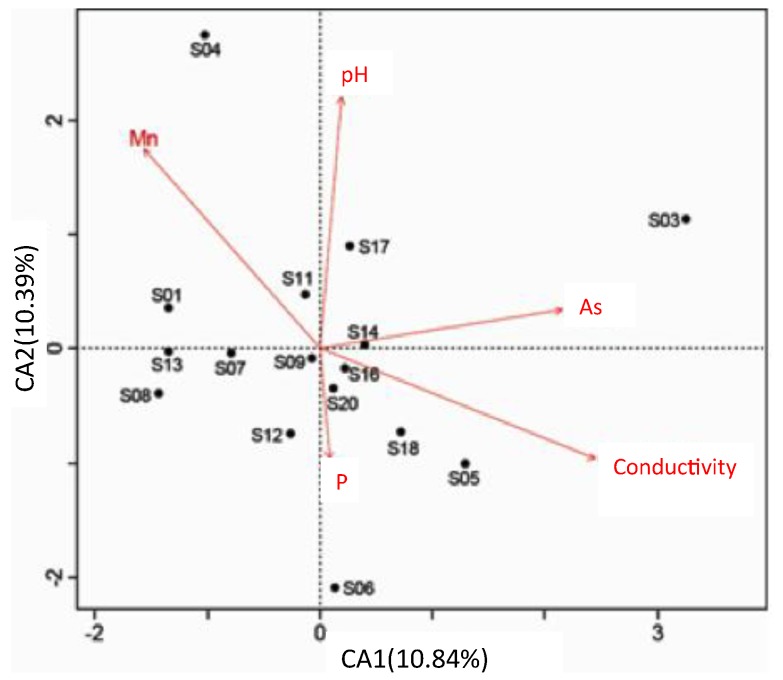
Canonical correspondence analysis of microbial data and the subset of environmental variables.

**Figure 4 ijerph-16-03068-f004:**
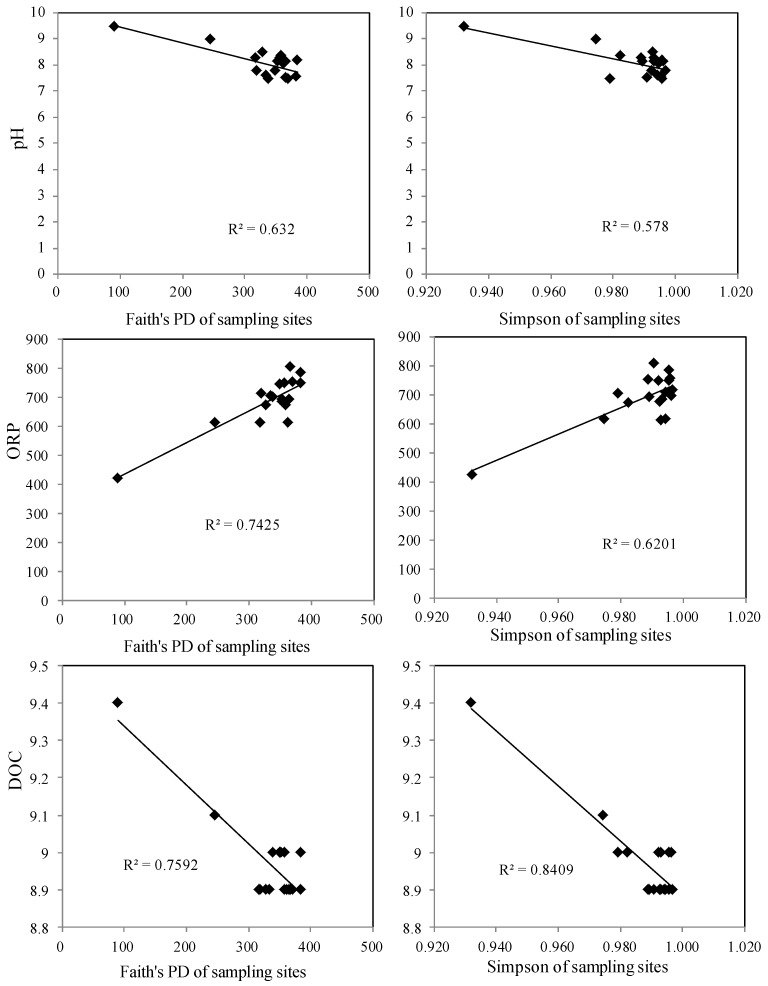
The simple correlation between the pH/ORP/DOC and Faith’s PD/Simpson values, respectively.

**Table 1 ijerph-16-03068-t001:** The physiochemical properties of overlying water and surface sediments.

Sites	Overlying Water								Surface Sediment					
	T (°C)	TN (μg/L)	TP (μg/L)	DO (mg/L)	ORP (mv)	EC (μs)	pH	SS (g/L)	DOC (mg/L)	Moisture Content (%)	DO (mg/L)	pH	ORP (mv)	H_2_S (mg/L)
1	25.2	1.59	57.1	14.18	514	514	9.20	3.32	8.8	55.54	7.31	8.10	405	0.185
2	22.6	2.87	79.6	6.75	641	584	9.21	5.25	9.1	59.28	6.26	9.43	418	0.191
3	21.7	1.94	49.3	7.12	627	591	8.89	4.25	8.9	44.26	6.02	8.21	441	0.174
4	20.6	1.84	55.4	13.87	582	442	8.71	4.05	8.7	48.43	6.79	8.33	423	0.190
5	21.2	1.71	67.2	13.29	747	601	7.41	3.35	9.1	59.06	7.02	7.56	454	0.207
6	20.0	1.89	66.7	14.59	609	594	7.42	3.87	8.9	59.15	7.82	8.02	470	0.205
7	18.8	1.91	64.3	13.90	1041	441	7.51	3.55	9.4	56.20	5.98	7.41	416	0.209
8	18.1	2.13	68.9	14.15	714	386	7.69	3.35	9.3	58.98	6.71	8.11	517	0.192
9	18.4	1.96	59.4	12.32	1047	541	7.70	1.98	8.9	53.29	6.88	7.45	507	0.194
10	17.2	1.89	43.5	9.76	784	571	7.90	3.30	9.1	33.86	5.31	8.92	468	0.187
11	18.0	2.07	42.9	12.01	871	383	7.91	2.70	8.6	47.49	5.78	8.22	441	0.175
12	21.3	1.97	41.2	13.12	890	387	7.92	2.95	9.0	27.34	6.51	7.72	432	0.166
13	19.1	2.31	51.4	12.59	765	347	7.83	1.15	8.8	49.72	6.09	7.43	467	0.165
14	18.9	2.41	43.5	11.97	779	380	7.93	3.22	8.9	41.88	6.41	7.53	445	0.190
15	21.7	2.11	53.6	12.38	1049	445	7.67	3.81	9.1	54.83	6.35	8.09	518	0.181
16	20.1	2.35	57.6	15.92	554	506	7.72	4.89	9.3	43.22	7.73	8.15	489	0.155
17	19.9	1.93	58.7	13.41	773	486	7.53	3.23	9.2	50.37	6.79	7.73	501	0.159
18	20.1	1.76	49.6	11.11	971	640	9.11	4.65	8.9	58.14	5.53	8.43	511	0.168

The standard deviations were all less than 5% of the corresponding values. T: temperature; TN: total nitrogen; TP: total phosphorus; DO: dissolved oxygen; EC: electrical conductivity; ORP: oxidation-reduction potential; DOC: dissolved oxygen concentration; SS: suspended solids.

**Table 2 ijerph-16-03068-t002:** The alpha diversity of the sampling sites in Lake Hongze.

Site	Shannon	Chao1	ACE	Faith’s PD	Number of OTUs	Simpson	Dominance
1	9.43	6348	6648	366	3916	0.989	0.011
2	5.51	596	593	90	577	0.932	0.068
3	9.05	5576	5826	359	3626	0.983	0.017
4	9.50	5168	5472	318	3462	0.993	0.007
5	9.50	5417	5577	335	3484	0.994	0.006
6	9.66	5882	6154	363	3730	0.995	0.005
7	9.51	6645	6827	358	3969	0.989	0.011
8	9.55	5741	5871	353	3607	0.993	0.007
9	9.56	6668	6676	367	3895	0.991	0.009
10	8.13	3697	3879	246	2420	0.975	0.025
11	8.90	5786	5988	339	3521	0.979	0.021
12	9.52	5853	6036	350	3709	0.992	0.008
13	9.95	6189	6460	370	3959	0.996	0.004
14	10.1	6882	7165	383	4281	0.996	0.004
15	9.93	5975	6217	353	3782	0.996	0.004
16	10.1	6751	6937	384	4201	0.996	0.004
17	9.93	4741	4916	320	3391	0.997	0.003
18	9.42	5626	5835	328	3477	0.993	0.007

ACE: adaptive communication environment; OTU: operational taxonomic unit; PD: phylogenetic diversity.

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
