# Peer review of "The Influence on Contaminant Bioavailability and Microbial Abundance of Lake Hongze by the South-to-North Water Diversion Project"

_ijerph, 2019, doi:10.3390/ijerph16173068_

Round 1

Reviewer 1 Report

I do not agree that the freshwater ecosytem is the most important water supply. This might apply for the region of your study, but not globally.

Line 33: are you talking about a safe water supply? Water security in this context is not the correct wording.

Line 94: microbial

Line 109: Why is Lake Hongze a typical freshwater lake? Typical for China? Typical for the region? Please provide more details.

Line 152 – 173: please provide the details in a table. This section is extremly hard to follow.

2.2.3: the statistical analyses is kept too short. More details needed to conlude wheather the right statistical tests were used. Is the data distrubited normally?

Table 1: Please highlight the differences in the text instead of just putting it in the table.

Figure 2: The figure is blurry and the axis designation cannot be read. Please provide a better quality of the figures.

Conclusion and Abstact: Need to be revised. State clearer the objective of the study and the lessons learnt from your work.

Author Response

Review1:

Comments and Suggestions for Authors

I do not agree that the freshwater ecosytem is the most important water supply. This might apply for the region of your study, but not globally.

Reply: Thanks for your suggestion. I have changed the descriptions in the revised manuscript as the following:

“The freshwater ecosystem represents the one of most important water supply for humans drinking and terrestrial plants irrigation” (Page 1, Line 31-32)

Line 33: are you talking about a safe water supply? Water security in this context is not the correct wording.

Reply: Thanks for your suggestion. I have changed the “water security” to “water quality safety” in the revised manuscript. (Page 1, Line 33)

Line 94: microbial

Reply: Thanks for your suggestion. I have changed the description to “microbial” in the revise manuscript.

Line 109: Why is Lake Hongze a typical freshwater lake? Typical for China? Typical for the region? Please provide more details.

Reply: Thanks for your suggestion. Hongze lake is fourth largest freshwater lake as well as the largest plain type reservoir in China, it plays an important role in agricultural economy and water supply in Jiangsu province. Additional, due to the widespread distribution of aquaculture and exogenous pollution import, lake Hongze is also in the extremely serious eutrophication. Hongze is the most largest lakes and with most important economic status in the South-to-North Water Diversion Project. Therefore, we speculate the lake Hongze might have the relatively complex influence of contaminants and microbial because of the water diversion. Consequently, we select Hongze Lake as the investigation area. To ensure the clarity of the manuscript, we have added the descriptions in the revised manuscript as follow:

“Hongze lake is fourth largest freshwater lake as well as the largest plain type reservoir in China, it plays an important role in agricultural economy and water supply in Jiangsu province [5]. The widespread distribution of aquaculture and exogenous pollution import resulted in the extremely serious eutrophication of Lake Hongze. Additional, Lake Hongze as a participant of the South-North Water Transfer Project, its contaminants internal loading are also thought to be affected by the water diversion; a hypothesis supported by previous ex-situ investigations [23]. Owing to its significant economic, geographic position, as well as the complex contaminants migration, Lake Hongze might be the typical freshwater lake for this investigation domain.” (Page 3, Line 110-117)

Line 152 – 173: please provide the details in a table. This section is extremely hard to follow.

Reply: Thanks for your suggestion. We have added the table to increase the readability in the revised manuscript. The details are showed as follow:

Table S1 Average mass in blank resin and measurement of detection limit of Hg, As, Fe, Mg, Cd, Pb and Mn.

FigS 3. The CID operation process of laible S and P.

2.2.3: the statistical analyses is kept too short. More details needed to conlude wheather the right statistical tests were used. Is the data distrubited normally?

Reply: Thanks for your suggestion. The F test and T test were applied to verify the difference between the labile concentrations obtained from different sampling sites. We have added the detailed descriptions in the revised Ms as follow:

“The 16S rRNA MiSeq sequencing data were operated with the modified pipelines of mothur and UPARSE (version 7.1 http://drive5.com/uparse/) [29]. The value obtained by the number of sequences belonging to specific taxon divided by the number of total sequences were used to represent the relative abundance (%) of individual taxa within each community. The alpha diversity (such as Shannon, Chao1, ACE, Faith’s Phylogenetic Diversity (PD), Simpson, and Dominance) and the beta-diversity (Bray-Curtis distance) were determined based on randomly selected sequences (the least sequences among the samples) from the sediment samples. To compare the species diversity between different samples, the rarefaction curve was constructed by sampling for all sequences. Then, the R program was applied to analyze the microbial community structure at different taxonomic levels [30].

All statistical analyses were performed using the SPSS statistical package (Version 18.0 for Windows) and some were implemented by R packages 
(http://www.r-project.org). The SPSS was applied for normality test of the physiochemical properties as well as the DGT measured labile concentrations at different depth/width of sampling sites, then, the t Test were applied to verify the difference. The relative abundances of specific lineages in different sampling sites were clustered by the R package g plots, the f and t test were also applied to verify the relative abundances of dominant lineages between the different sampling sites. Furthermore, the community dissimilarities (Bray−Curtis distances) among the different sampling sites in Lake Hongze were significant (P < 0.05). To visualize the differences in microbial communities between all sampling sites of Lake Hongze, the diversity indices (Phylotypes and Faith’s PD) of different sampling sites were exhibited. The “BioEnv” procedure and the canonical correspondence analysis (CCA) were used to select the best environmental variables for the microbial communities of all sampling sites in Lake Hongze [31].”(Page 6, Line 220-232)

Table 1: Please highlight the differences in the text instead of just putting it in the table.

Reply: Thanks for your suggestion. I have highlight the differences of the table in bold font.

Figure 2: The figure is blurry and the axis designation cannot be read. Please provide a better quality of the figures.

Reply: Thanks for your suggestion. I have changed the figure in clearer version.

Conclusion and Abstact: Need to be revised. State clearer the objective of the study and the lessons learnt from your work.

Reply: Thanks for your suggestion. We have revised the “Conclusion and Abstact” in the revised manuscript to increase the readability.

Reviewer 2 Report

General comments:

The authors unfold the linkage between typical microbial structure and elementary pollution conditions in Lake Hongzhe by high-throughput pyrosequencing and DGT. The methods and results are mostly acceptable, while the influence by water diversion should be further specified and explained. In addition, the writing style need much more improvement.

Specific comments:

Title: It should be more specified. "contaminated", "community conditions" were too general.

Abstract

L 11-16: Such general information is a waste of words in Abstract. The methods did not mention how to identify the influence of water transfer project.

L 20-26: More specific results should be added, other than the general evaluations.

L26-28: This conclusion was beyond your study and such statements decreased the value of this arduous study.

L 23-25: The test or calculation method of the sequestration rates should be added concisely.

L 41-42: Please cite the original reference.

Introduction

What about the reviews focused on the influences of water transfer project?

L 65-93: Such a big section only used for DGT technology description was unnecessary.

L 108-113, 116-121: Such sentences should be moved to the Materials and Methods.

L113-116: I think this sentence was the objective. Were the objective of "migration, distribution and predication" achieved in this study?

Materials and Methods

L 129-130: Please give the references or evidences that why such sites were defined as the "obviously affected area"? It's very important to justify the method used in this study to identify or differentiate the influence of water transfer project.

Fig. 1: It's a serious mistake of inciting China's map incorrectly. It should be rejected if not considering the merits of this study.

Results and discussion

This section was difficult to read logically. Results and discussion should be divided into two sections.

L224-225: Move to "M & M" section.

Table 1: Any abbreviations should be fully noted in the table captions. Which items belong to overlying water? Why all the values without S.E. or S.D.?

Fig. 2: The differences between different elements were too small to differentiate them rapidly.

Table 2: What the difference between "ID S01" and "site 1" in Table 1?

Fig. 4: What about the significances of all the fitting equations?

Conclusions

Too simple to conclude the main highlights of this study. Any general known or predictable  points should be deleted.

Author Response

Review 2:

Comments and Suggestions for Authors

General comments:

The authors unfold the linkage between typical microbial structure and elementary pollution conditions in Lake Hongzhe by high-throughput pyrosequencing and DGT. The methods and results are mostly acceptable, while the influence by water diversion should be further specified and explained. In addition, the writing style need much more improvement.

Reply: Thanks for your suggestion. We have added the references and descriptions of the influence on labile fractions and microbial community by the water diversion. Moreover, we have increased the readability by the polish company.

Specific comments:

Title: It should be more specified. "contaminated", "community conditions" were too general.

Reply: Thanks for your suggestion. The title was changed to “The Influence on Contaminants Bioavailability and Microbial abundance of Lake Hongze By the South-to-North Water Diversion Project” in the revised manuscript.

Abstract

L 11-16: Such general information is a waste of words in Abstract. The methods did not mention how to identify the influence of water transfer project.

Reply: Thanks for your suggestion. I have changed the descriptions in the revised manuscript as follow:

“South-North Water Transfer Project as world famous Project was built to alleviate serious water shortages of northern China. Considering the lake Hongze is an important freshwater lake in this region, analyzing water diversion influence on typical contaminants bioavaiability and microbial abundance could establish the overall understanding of hydrodynamic variation.” (Page 1, Line 11-14)

L 20-26: More specific results should be added, other than the general evaluations.

Reply: Thanks for your suggestion. We have added the specific results descriptions in the revised abstract of manuscript as follow:

“The pH, Conductivity, labile Mn、As and P were shown to be the primary environmental factors affecting the abundance and diversity of microbial. Except the bioturbation affected sites were controlled by labile Mn and pH, sites distributed in the water diversion area most affected by As and conductivity, indicated little spatial discrepancy. Furthermore, sites 2 with higher bioturbation abundance and site 10 with more strong hydrodynamics were with low alpha diversity compared to the other sites. Consequently, the typical contaminants bioavailability such as P, S, As, Hg, Fe, Mg, Cd, Pb, and Mn as well as the diversity and abundance of microbial in the sites influence by the water diversion have the significantly difference to the other sites. Thus, the South-North Water Transfer Project impact on participant lakes were non-negligible for the overall investigation. ” (Page1, Line 20-29)

L26-28: This conclusion was beyond your study and such statements decreased the value of this arduous study.

Reply: Thanks for your suggestion. I have changed the descriptions in the revised manuscript. (Page 1, Line 28-29)

L 23-25: The test or calculation method of the sequestration rates should be added concisely.

Reply: Thanks for your suggestion. The extraction has been added in the abstract and the M&M of the revised manuscript. The revised abstract was shown as follow:

“The DGT could effective obtained more than the 85% bioavailable concentrations of the corresponding contaminant, the results showed that labile P, S, Fe, As, and Hg concentrations were higher in areas influenced by water transfer.” (Page 1, Line 18-20)

L 41-42: Please cite the original reference.

Reply: Thanks for your suggestion. I have added the original reference in the revised manuscript. The added reference was listed as follow:

“Sun, W., Xiao, T., Sun, M., et al. Diversity of the Sediment Microbial Community in the Aha Watershed (Southwest China) in Response to Acid Mine Drainage Pollution Gradients [J]. Applied & Environmental Microbiology, 2015, 81 (15): 4874-4884.”

Introduction

What about the reviews focused on the influences of water transfer project?

Reply: Thanks for your suggestion. The water transfer project is necessary to alleviate serious water shortages of northern China, its obviously positive effects cannot be ignored. However, the contaminants and microbial will trend to convergence or stabilization, it might be the research emphasis. Consequently, I attempted to illustrate the influence of the water diversion on the typical contaminants and microbial diversity in the freshwater lake.

L 65-93: Such a big section only used for DGT technology description was unnecessary.

Reply: Thanks for your suggestion. I have reduced the descriptions of DGT in the revised manuscript.

L 108-113, 116-121: Such sentences should be moved to the Materials and Methods.

Reply: Thanks for your suggestion. I have moved the descriptions to the Materials and Methods. The descriptions of L 108-113 were moved to Line 113-119, The descriptions of 116-121 were moved to Line 131-136.

L113-116: I think this sentence was the objective. Were the objective of "migration, distribution and predication" achieved in this study?

Reply: Thanks for your suggestion. We selected the typical contaminants i.e., Hg, As, Fe, Mg, Cd, P, and S as the object, aiming to obtain their bioavailable fractions distribution and migration as well as their influence factor, in the different width and depth of sampling sites. Considering the abundance and the diversity of the microbial could not only be the dependent variable of contaminants also the indicator for the ecological. Consequently, we attempted to obtain the main influence such as bioturbation,hydrodynamics of different area through the contaminants and microbial diversity information. Thus, the comparison of the target elements variation in different area could be the evidence caused by water diversion. To sum up, we have reached the purpose of investigation objective.

Materials and Methods

L 129-130: Please give the references or evidences that why such sites were defined as the "obviously affected area"? It's very important to justify the method used in this study to identify or differentiate the influence of water transfer project.

Reply: Thanks for your suggestion. We have added the reference to add the persuasiveness of the revised manuscript. The added reference was listed as the follow:

“ Yu H, Zhang W B, Lu S Y, et al. Spatial Distribution Characteristics of Surface Sediments Nutrients in Lake Hongze and Their Pollution Status Evaluation (in chinese)[J]. Environmental Science, 2010, 31 (4): 961-968.”

Fig. 1: It's a serious mistake of inciting China's map incorrectly. It should be rejected if not considering the merits of this study.

Reply: Thanks for your suggestion. Due to the size constraint, we have deleted the China's map in the revised manuscript.

Results and discussion

This section was difficult to read logically. Results and discussion should be divided into two sections.

Reply: Thanks for your suggestion. We have divided the “Results and discussion” into two sections in the revised manuscript.

L224-225: Move to "M & M" section.

Reply: Thanks for your suggestion. We have moved the descriptions to the "M & M" section in the revised manuscript. (Page 4, Line 143-144)

Table 1: Any abbreviations should be fully noted in the table captions. Which items belong to overlying water? Why all the values without S.E. or S.D.?

Reply: Thanks for your suggestion. We have added the fully noted and de S.D in the revised manuscript.

Fig. 2: The differences between different elements were too small to differentiate them rapidly.

Reply: Thanks for your suggestion. Owing to spatial confined, we must put all the contaminants information in one figures. However, to increase the readability, we have added the size of the figures.

Table 2: What the difference between "ID S01" and "site 1" in Table 1?

Reply: Thanks for your suggestion. We have unified the description in tables of the revised manuscript.

Fig. 4: What about the significances of all the fitting equations?

Reply: Thanks for your suggestion. We attempted to obtain the main influence on the abundance of microbial variations. Hence, we selected the typical physicochemical properties to search the relationship between the Fiath’s PD/Simpson.

Conclusions

Too simple to conclude the main highlights of this study. Any general known or predictable points should be deleted.

Reply: Thanks for your suggestion. We have revised the descriptions in the manuscript to increase readability.

Reviewer 3 Report

The manuscript “The Influence on Contaminated and microbial community conditions of Lake Hongze By the South-to-North Water Diversion Project” is an interesting research article addressing the ecologically relevant topic of contaminants and microbial diversity in freshwater altered by a water transfer project in China. I believe that the authors did good scientific work during this study, however the English needs serious revision to make the manuscript easily readable and understandable. In addition, I detected the following specific issues:

Line 60 – A reference for this statement is needed.

Line 74 – Could the authors explain what is a “pre-concentration”?

Line 94 and following – Several spelling mistakes

Lines 132-133 – I do not see the connection of this sentence with the rest of the paragraph.

Line 171 – Correct the reference formatting error.

Lines 188-189 – The reference for the primers used is missing.

Lines 189-190 – Please clarify if the primer is present in both or only the reverse primer.

Line 192 – The authors refer to pyrosequencing before, but they seem to use Illumina technology instead. Please fix this issue through the text.

Line 199 – Why did the authors choose this overlap length? It seems quite short.

Line 201 – An appropriate reference for mothur and usearch is missing.

Line 204 – Please define PD in “Faith’s PD”. Also clarify the taxonomic level considered for the richness measurement used (phylotypes).

Lines 204-205 – Please separate alpha and beta-diversity indices to make this sentence clearer.

Line 206 – What do the authors mean by “randomly selected sequences from the samples”?

Lines 211-212 – Clarify which specific analyses were performed using each software.

Lines 216-217 – This is a result, please move to the appropriate place.

Lines 217-219 – Clarify how the graphical results were obtained.

Lines 220-221 – Explain this better.

Lines 224-227 – All this part seems to belong to methods. Please relocate accordingly.

Table 1 – Maybe the authors want to include in the table also the standard deviation. It would be useful to have a similar table for the metals concentration, to avoid presenting a long list of numbers for each of them in the text.

Line 324 – Some indices presented here were not described in methods. Please be congruent throughout the text.

Lines 327-328 – Could the authors explain what do they mean by “random selection of sampling sites”? They did not analyze all of the sites for the alpha-diversity analysis?

Lines 333-335 – Please verify the primers used are adequate to describe both Archaea and Bacteria. If they are not, this could explain the low representation of Archaea in the samples.

Lines 354-356 – Please format all the genera and species names in italics.

Lines 381-385 – Several misspellings for “Faith’s”.

Line 402 – Higher or lower? And if there are opposite results in different sites, please specify them.

Author Response

Review3:

Comments and Suggestions for Authors

The manuscript “The Influence on Contaminated and microbial community conditions of Lake Hongze By the South-to-North Water Diversion Project” is an interesting research article addressing the ecologically relevant topic of contaminants and microbial diversity in freshwater altered by a water transfer project in China. I believe that the authors did good scientific work during this study, however the English needs serious revision to make the manuscript easily readable and understandable.

Reply: Thanks for your suggestion. We have increased the readability by the polish company.

Line 60 – A reference for this statement is needed.

Reply: Thanks for your suggestion. We have added the related reference in the revised manuscript.

Line 74 – Could the authors explain what is a “pre-concentration”?

Reply: Thanks for your suggestion. The device could accumulate the target elements increased with the deploy time increase under the circumstance of resin gel absorptive capacity unsaturation. Consequently, DGT device could obtain more accuracy contamination information by deploy time extension. Thus, we thought this characteristic could be defined as “pre-concentration”.

Line 94 and following – Several spelling mistakes

Reply: Thanks for your suggestion. I have corrected the spelling mistakes in the revised manuscript.

Lines 132-133 – I do not see the connection of this sentence with the rest of the paragraph.

Reply: Thanks for your suggestion. We attempted to illustrate the distribution of the different sampling sites. Consequently, we did not delete the related descriptions. However, to increase the readability, we have changed the descriptions in the revised manuscript.

Line 171 – Correct the reference formatting error.

Reply: Thanks for your suggestion. We have corrected the reference in the revised manuscript.

Lines 188-189 – The reference for the primers used is missing.

Reply: Thanks for your suggestion. We have added the reference for the primers in the revised manuscript as follow:

“Loman, N J., Misra, R V., Dallman, T J., et al. Corrigendum: Performance comparison of benchtop high-throughput sequencing platforms [J]. Nature Biotechnology, 2012, 30 (5): 434-9.”

Lines 189-190 – Please clarify if the primer is present in both or only the reverse primer.

Reply: In this study, the V4–V5 hypervariable regions of 16S rRNA were PCR amplified from microbial genomic DNA using the following universal primers: V515F, 5’-GTGCCAGCMGCCGCGG-3’; V907R, 5’-CCGTCAATTCMTTTRAGTTT-3’. This information has been given in the revised manuscript at Line 191-193.

Line 192 – The authors refer to pyrosequencing before, but they seem to use Illumina technology instead. Please fix this issue through the text.

Reply: In this study, high-throughput sequencing technique was used to investigate the microbial communities on an Illumina MiSeq platform. The word “pyrosequencing” has been removed and the pharse “high-throughput sequencing” was used through the text.

Line 199 – Why did the authors choose this overlap length? It seems quite short.

Reply: In this study, the resulting sequences were screened and filtered for quality and length. Sequences with a length shorter than 50 bp, having more than two primer mismatches were removed, and paired end reads were merged accordingly (Caporaso et al., 2010). This information has been given in the revised manuscript at Line 196-198.

Line 201 – An appropriate reference for mothur and usearch is missing.

Reply: Thanks for your suggestion. The 16S rRNA MiSeq sequencing data were operated with the modified pipelines of mothur and UPARSE (version 7.1 http://drive5.com/uparse/) (Caporaso et al., 2010). We have added the reference in the revised manuscript.

Line 204 – Please define PD in “Faith’s PD”. Also clarify the taxonomic level considered for the richness measurement used (phylotypes).

Reply: Thanks for your suggestion. In this study, PD is the abbreviation of phylogenetic diversity, and this information has been given in the revised manuscript at Line 203-204. The richness of microbial communities was determined based on the value of Chao1, ACE, Faith’s PD.

Lines 204-205 – Please separate alpha and beta-diversity indices to make this sentence clearer.

Reply: In this study, the alpha diversity included Shannon, Chao1, ACE, Faith’s Phylogenetic Diversity (PD), Simpson, and Dominance and the Bray-Curtis distance was determined to represent beta-diversity. All the necessary modifications have been made in the revised manuscript at Line 202-205.

Line 206 – What do the authors mean by “randomly selected sequences from the samples”?

Reply: In this study, the alpha diversity and the beta-diversity were determined based on randomly selected sequences (the least sequences among the samples) from the sediment samples. This information has been given in the revised manuscript at Line 202-205.

Lines 211-212 – Clarify which specific analyses were performed using each software.

Reply: In this study, all statistical analyses were performed using the SPSS statistical package (Version 18.0 for Windows) and some were implemented by R packages (http://www.r-project.org). Specifically, the SPSS was applied for normality test of the physiochemical properties as well as the DGT measured labile concentrations at different depth/width of sampling sites, then, the t Test were applied to verify the difference. The relative abundances of specific lineages in different sampling sites were clustered by the R package g plots, the f and t test were also applied to verify the relative abundances of dominant lineages between the different sampling sites. Furthermore, the community dissimilarities among the different sampling sites in Lake Hongze were determined based on Bray−Curtis distances. These revision have been made in the text at line 210-218.

Lines 216-217 – This is a result, please move to the appropriate place.

Reply: Thanks for your suggestion. The expression has been revised as “Furthermore, the community dissimilarities among the different sampling sites in Lake Hongze were determined based on Bray−Curtis distances.” Please check at line 216-218.

Lines 217-219 – Clarify how the graphical results were obtained.

Reply: In this study, the alpha diversity (such as Shannon, Chao1, ACE, Faith’s PD, Simpson, and Dominance) were determined and given in Table 2. Please check.

Lines 220-221 – Explain this better.

Reply: In this study, abiotic factors influencing the community were evaluated using a BioENV procedure, and canonical correspondence analysis (CCA) were used to select the best environmental variables for the microbial communities of all sampling sites in Lake Hongze. These revision have been made in the text at Line 219-221.

Lines 224-227 – All this part seems to belong to methods. Please relocate accordingly.

Reply: Thanks for your suggestion. We have moved the descriptions to the methods section in the revised manuscript.

Table 1 – Maybe the authors want to include in the table also the standard deviation. It would be useful to have a similar table for the metals concentration, to avoid presenting a long list of numbers for each of them in the text.

Reply: Thanks for your suggestion. Due to the variations of the target elements were complex along the depth and width, we chose the figures to exhibit. Furthermore, we have added the description to illustrate the standard deviations of the corresponding values in the Table 1.

Line 324 – Some indices presented here were not described in methods. Please be congruent throughout the text.

Reply: In this study, the alpha diversity included Shannon, Chao1, ACE, Faith’s Phylogenetic Diversity (PD), Simpson, and Dominance. This information has been added in the section of Materials and Methods, and all the necessary modifications have been made throughout the text.

Lines 327-328 – Could the authors explain what do they mean by “random selection of sampling sites”? They did not analyze all of the sites for the alpha-diversity analysis?

Reply: Thanks for your suggestion. We have deleted this sentence to increase the readability in the revised manuscript.

Lines 333-335 – Please verify the primers used are adequate to describe both Archaea and Bacteria. If they are not, this could explain the low representation of Archaea in the samples.

Reply: Thanks for your suggestion. In this study, the V4–V5 hypervariable regions of 16S rRNA were PCR amplified from microbial genomic DNA using the following universal primers: V515F, 5’-GTGCCAGCMGCCGCGG-3’; V907R, 5’-CCGTCAATTCMTTTRAGTTT-3’. Accordingly, this regions can be used to identify most eubacteria and archaebacteria, were often selected to determine the bacterial communities in environmental samples (Liao et al., 2019). Also, some archaea can be identified using this hypervariable regions, and the obtained results were used to compared with the results of bacteria.

Liao, H., Yu, K., Duan, Y., Ning, Z., Li, B., He, L., & Liu, C. (2019). Profiling microbial communities in a watershed undergoing intensive anthropogenic activities. Science of The Total Environment, 647, 1137-1147.

Lines 354-356 – Please format all the genera and species names in italics.

Reply: Thanks for your suggestion. We have format all the genera and species names in italics in the revised manuscript.

Lines 381-385 – Several misspellings for “Faith’s”.

Reply: Thanks for your suggestion. We have corrected the misspellings in the revised manuscript.

Line 402 – Higher or lower? And if there are opposite results in different sites, please specify them.

Reply: Thanks for your suggestion. We have changed the descriptions in the conclusion of the revised manuscript. (Page 14, Line 404-406)

Round 2

Reviewer 1 Report

The quality of the figures is still not satisfying. The labeling of the axis cannot be read. Please provide a better quality of all figures.

Author Response

Review 1

The quality of the figures is still not satisfying. The labeling of the axis cannot be read. Please provide a better quality of all figures.

Reply: Thanks for your suggestion. We have changed the labeling of the axis to increase the readability. Moreover, the quality of the figure has also been modified to satisfy the request.

Reviewer 2 Report

Most comments were responded basically. However, my major concern that how and why to classify the 18 sites into 3 categories to compare the differences caused by water diversion project, did not replied correctly. The author added a reference published in 2010 (“Yu H, Zhang W B, Lu S Y, et al. Spatial Distribution Characteristics of Surface Sediments Nutrients in Lake Hongze and Their Pollution Status Evaluation (in chinese)[J]. Environmental Science, 2010, 31 (4): 961-968.”), while the South-to-North Water Diversion Project was implemented in 2013. In addition, such reference was not added in the reference list.

Author Response

Review2

Most comments were responded basically. However, my major concern that how and why to classify the 18 sites into 3 categories to compare the differences caused by water diversion project, did not replied correctly. The author added a reference published in 2010 (“Yu H, Zhang W B, Lu S Y, et al. Spatial Distribution Characteristics of Surface Sediments Nutrients in Lake Hongze and Their Pollution Status Evaluation (in chinese)[J]. Environmental Science, 2010, 31 (4): 961-968.”), while the South-to-North Water Diversion Project was implemented in 2013. In addition, such reference was not added in the reference list.

Reply: Thanks for you suggestion. We have added the reference in the reference list of the revised manuscript. We chose this reference aiming to illustrate the nutrients distributions of the surface sediment without the influence of the water diversion. Moreover, as the figure 1 shown, the sampling sites distributed near the inlet or outlet of the lake were usually identified as water diversion affected sites. Furthermore, to increase the power of persuasion, the hydrodynamic model will be conceived in our further research.
